# The inflammatory response of human pancreatic cancer samples compared to normal controls

**Kathryn J. Brayer[1], Joshua A. Hanson[2], Shashank Cingam[3], Cathleen Martinez[2], Scott A. Ness[1], Ian Rabinowitz** [3] *

**1** Department of Internal Medicine, Molecular Medicine, University of New Mexico, Albuquerque, New Mexico, United States of America, **2** Department of Pathology, University of New Mexico, Albuquerque, New Mexico, United States of America, **3** Division of Hematology- Oncology, Department of Internal Medicine, University of New Mexico, Albuquerque, New Mexico, United States of America

* irabinowitz@salud.unm.edu

**Data Availability Statement:** All relevant data are within the paper and its Supporting information files. Genomics Shared Resource at the UNM Comprehensive Cancer Center. RNA sequencing

## Abstract

Pancreatic ductal adenocarcinoma (PDAC) is a poor prognosis cancer with an aggressive growth profile that is often diagnosed at late stage and that has few curative or therapeutic options. PDAC growth has been linked to alterations in the pancreas microbiome, which could include the presence of the fungus *Malassezia*. We used RNA-sequencing to compare 14 matched tumor and normal (tumor adjacent) pancreatic cancer samples and found *Malassezia* RNA in both the PDAC and normal tissues. Although the presence of *Malassezia* was not correlated with tumor growth, a set of immune- and inflammatory-related genes were up-regulated in the PDAC compared to the normal samples, suggesting that they are involved in tumor progression. Gene set enrichment analysis suggests that activation of the complement cascade pathway and inflammation could be involved in pro PDAC growth.

## Introduction

Pancreatic cancer (PDAC) is the 9th most common cancer in the US but is the 4th most common cause of cancer related death (~54,000/year and ~44,000/year respectively). The median 5 year survival for stage 4 disease is 9% [1]. The high death rate with respect to the prevalence rate is due to poor early detection and a lack of meaningful advancement in systemic therapeutics. The most common somatic mutations, Kirsten rat sarcoma viral oncogene (*KRAS*), tumor protein p53 (*TP53*), cyclin dependent kinase inhibitor 2 A (*CDKN2A*), and SMAD family member 4 (*SMAD4*) have been identified by whole-exome and -genome sequencing of large PDAC cohorts and form the majority of unique mutations in patients with PDAC [2].

The tumor microenvironment (TME), which represents a complex ecosystem involving interactions between immune cells, cancer cells, stromal cells, and the extracellular matrix, can support tumor proliferation, survival, and metastasis and can be highly immunosuppressive [3–5].

In one paper, they found in patients with a resected PDAC and a higher tumor microbial diversity did better than resected PDAC with a lower microbial diversity. They also showed

data is available for download from the NCBI BioProject database using study accession number PRJNA940178 (https://www.ncbi.nlm.nih.gov/bioproject/?term=PRJNA940178).

**Funding:** This research was partially supported by the University of New Mexico Comprehensive Cancer Center Support Grant NCI University of New Mexico Cancer Research & Treatment Center P30CA118100(IR) https://unmhealth.org/cancer/research/sharedresources/ and the Analytical and Translational Genomics Shared Resource, which receives additional support for the State of New Mexico.(SN,KB) R01DE023222 https://app.dimensions.ai/details/grant/grant.2491524, U2CCA252973 https://www.ncbi.nlm.nih.gov/pmc/articles/PMC9223088/(SN,KB) and University of New Mexico Comprehensive Cancer Center Support Grant NCI University of New Mexico Cancer Research & Treatment Center P30CA118100(SN,KB) https://unmhealth.org/cancer/research/shared-resources/ and by Department of Defence grant: Defence Threat Reduction Agency https://www.dtra.mil/HDTRA12010015.(SN,KB). The funders had no role in study design, data collection and analysis, decision to publish, or preparation of the manuscript.

**Competing interests:** The authors have declared that no competing interests exist.

that patients who carried the long term survival (LTS) microbiome signature, namely (Pseudoxanthomonas, Streptomyces, and Saccharopolyspora) lived a long time. There was no microbiome signature for the patients who had a short term survival (STS). Patients with high diversity and low diversity had overall survival of 9.66 vs. 1.66 years respectively using univariate Cox proportional hazard models.

They presented data in a mice model showing the benefit of a fecal microbiota transplantation of the LTS compared to the STS gut microbiome. Thus, tumor micro-environment is an important part of the prognosis of patients with resected PDAC [6].

On the other hand, bacterial ablation in the pancreas was shown to alter the immune system, allowing the immune system response to decrease growth of the tumor by a reduction in myeloid-derived suppressor cells and an increase in M1 macrophage differentiation, promoting Th1 differentiation of CD4+ T cells and CD8+ T cell activation. Bacterial ablation also enabled efficacy for checkpoint-targeted immunotherapy by up-regulating programmed death (PD-1) expression [7]. More recently the same group looked at the role of the fungal microbiome in PDAC [8]. In a mouse model they showed a significant migration of fungi from the lumen of the gut into the lumen of the pancreas in both mice and humans with PDAC compared to non-malignant controls. Both in mice and humans, the *Malassezia* fungus was highly enriched in the malignant pancreatic tumor. They demonstrated that the binding of mannose-binding lectin (MBL) to glycans of the fungal wall, and lectin pathway activation was required for oncogenic progression in pancreatic cancer.

Furthermore, deletion of MBL or C3 in the extra-tumoral compartment or knockdown of C3ar in tumor cells were both protective against tumor growth in mouse models. There was thus a suggestion that using anti-fungal agents at some point of the treatment of pancreatic cancer may result in a shrinking or slowing down of the growth of the tumor. In a paper reviewing expression of complement in various cancers most cancers over-expressed C3 but neutrally-expressed C5, and the most over expressed gene was CD59, suggesting efficient protection of malignant cells from complement- mediated killing [9].

To document the expression of C3 and *Malassezia* and associated genes in our patient population, we decided to perform studies on paraffin embedded pancreatic cancer biopsies using pancreatic cancer and their normal tissue counterpart samples from the University of New Mexico (UNM) Tissue Repository. We obtained exemption to perform the study from the local Human Research Protections Program.

## Results

### Microbiome results

Using optimized methods [10], we used RNA-seq analysis on matched tumor-normal PDAC tissue samples derived from formalin-fixed paraffin embedded (FFPE) slices for 15 patients obtained from UNM Tissue Repository, generating high quality data for 14 patients, with an average of 32 x 10^-6 reads per sample (Tables 1 and 2). After sequencing, reads were taxonomically classified using kraken2 [11–13] and braken [14] against a library containing human, viral, bacterial, and fungal genomes, including the genomes of several *Malassezia* species. The fungus *Malassezia* was present in all tumor samples at varying concentrations (Tables 1 and 2). We also detected *Malassezia* in all of the normal tissue samples, which is in contrast to a previous study (Tables 1 and 2) [8]. Despite reports that PDAC tumors have a lower microbial diversity than normal pancreatic tissues [6], we found no significant difference in the number of microbial genera present in normal samples versus tumor samples (Table 1).

**Table 1. RNA sequencing statistics.**

| | No. Samples | Mean Total Reads, x10^-6 (range) | Mean Human Reads, x10^-6 (range) | Mean Number of Genera (range) | Mean Percent Reads Malassezia (range) |
|---|---|---|---|---|---|
| **Normal Samples** | 14 | 27.0 (14.4–40.0) | 13.0 (6.9–19.4) | 220 (74–621) | 0.09 (0.005–0.710) |
| **Tumor Samples** | 14 | 36.6 (12.2–96.4) | 17.7 (5.9–46.6) | 224 (71–648) | 0.02 (0.003–0.054) |

## Transcriptome analysis

Unsupervised multidimensional scaling (MDS) was used to compare the normal and tumor samples. The MDS plot (Fig 1A) revealed two tumor sample outliers, which were more similar to the normal samples than the other tumor samples. In addition, most of the normals clustered tightly together, while the tumor samples were more dispersed, indicating that the tumor gene expression pattern were more heterogeneous while the normals were more homogenous. Differential gene expression (DGE) analysis of the 14 matched tumor-normal patient samples

**Table 2. RNA sequencing statistics by sample.**

| Sample Name | Tissue | Malassezia | Total Reads | Total Reads Aligning to hg38 | Percent of Total Reads Aligning to Malassezia |
|---|---|---|---|---|---|
| N_1 | Normal | yes | 18858261 | 9085011 | 0.011 |
| T_1 | Tumor | yes | 22821514 | 10788345 | 0.01 |
| N_2 | Normal | yes | 34870246 | 16274139 | 0.178 |
| T_2 | Tumor | yes | 28604529 | 13761918 | 0.025 |
| N_3 | Normal | yes | 20067601 | 9752758 | 0.006 |
| T_3 | Tumor | yes | 25180357 | 12250509 | 0.003 |
| N_4 | Normal | yes | 17769109 | 8488343 | 0.005 |
| T_4 | Tumor | yes | 39754255 | 19063222 | 0.011 |
| N_5 | Normal | yes | 29585656 | 14189749 | 0.071 |
| T_5 | Tumor | yes | 36737424 | 17795862 | 0.009 |
| N_6 | Normal | yes | 34809441 | 16864905 | 0.012 |
| T_6 | Tumor | yes | 24993439 | 12139287 | 0.024 |
| N_7 | Normal | yes | 35439899 | 17087723 | 0.011 |
| T_7 | Tumor | yes | 30130953 | 14606463 | 0.011 |
| N_9 | Normal | yes | 40309681 | 19442161 | 0.031 |
| T_9 | Tumor | yes | 24666781 | 12105195 | 0.012 |
| N_10 | Normal | yes | 23369068 | 11433849 | 0.018 |
| T_10 | Tumor | yes | 33765469 | 16437724 | 0.021 |
| N_11 | Normal | yes | 14461851 | 6960809 | 0.046 |
| T_11 | Tumor | yes | 12230195 | 5977344 | 0.02 |
| N_12 | Normal | yes | 19113684 | 9313691 | 0.032 |
| T_12 | Tumor | yes | 65797247 | 31684330 | 0.054 |
| N_13 | Normal | yes | 22638160 | 11090895 | 0.102 |
| T_13 | Tumor | yes | 29198837 | 14100141 | 0.027 |
| N_14 | Normal | yes | 27131048 | 13192198 | 0.021 |
| T_14 | Tumor | yes | 42312723 | 20659150 | 0.034 |
| N_15 | Normal | yes | 38754908 | 18678438 | 0.009 |
| T_15 | Tumor | yes | 96452644 | 46639714 | 0.003 |

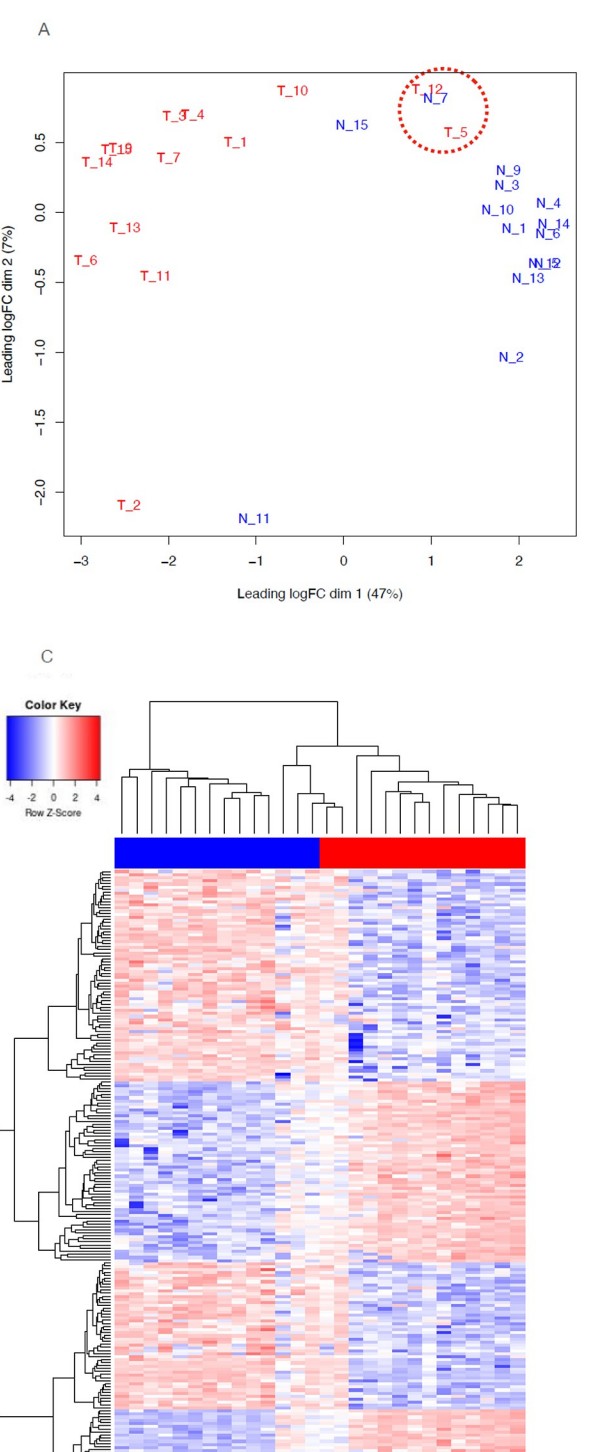

**Fig 1.** (A) Multidimensional scale plot of PDAC tumor (red bar) and normal (blue bar) RNA-seq data. (B) Volcano plot summarizing the differential gene expression analysis, showing log2 fold change vs. log10 of the p-value (BH adjusted). Red indicates genes that were > = 1.5 log2 fold change and had an adjusted p-value < = 0.05. (C) The heatmap summarizes the unsupervised clustering and differential gene expression analysis comparing the normal samples to the tumor samples. Dendrograms show the unsupervised hierarchical clustering of samples (top) and genes (left side). A larger version of this heatmap with samples and genes labeled is provided in the supplementary results (S1 Fig in S1 Appendix).

**Table 3. Select gene set enrichment analysis results.** Cluster profiler was used to probe genesets at the molecular signature. Database using gene differentially expressed between tumor and normal samples (Yu et al 2012, Subramanian et al 2005, Liberzon et al 2015).

| Description | Geneset | GeneRatio | BgRatio | Adjusted pValue (method = BH) | Qvalue |
|---|---|---|---|---|---|
| GRUETZMANN_PANCREATIC_CANCER_UP | C2: curated gene sets, Canonical pathways subset | 87/1050 | 357/ 21697 | 2.31897E-34 | 1.51190E-34 |
| GRUETZMANN_PANCREATIC_CANCER_DN | C2: curated gene sets, Canonical pathways subset | 65/1050 | 199/ 21697 | 1.12991E-33 | 7.36667E-34 |
| HALLMARK_EPITHELIAL_MESENCHYMAL_TRANSITION | H: hallmark gene sets | 80/452 | 200/4383 | 2.26022E-28 | 1.66543E-28 |
| KEGG_COMPLEMENT_AND_COAGULATION_CASCADES | C2: curated gene sets, Canonical pathways subset | 17/1050 | 69/21697 | 4.41776E-07 | 2.88024E-07 |
| HALLMARK_INTERFERON_GAMMA_RESPONSE | H: hallmark gene sets | 39/452 | 200/4383 | 0.00081 | 0.00060 |
| GOBP_COMPLEMENT_ACTIVATION | C5: ontology gene sets, GO:BP subset | 22/999 | 171/ 19303 | 0.00167 | 0.00128 |
| GOBP_REGULATION_OF_COMPLEMENT_ACTIVATION | C5: ontology gene sets, GO:BP subset | 17/999 | 114/ 19303 | 0.00173 | 0.00132 |
| HALLMARK_ANGIOGENESIS | H: hallmark gene sets | 12/452 | 36/4383 | 0.00193 | 0.00143 |
| HP_ABNORMALITY_OF_COMPLEMENT_SYSTEM | C5: ontology gene sets, HPO subset | 8/999 | 31/19303 | 0.00256 | 0.00195 |
| HALLMARK_KRAS_SIGNALING_UP | H: hallmark gene sets | 35/452 | 200/4383 | 0.00853 | 0.00629 |
| HALLMARK_INFLAMMATORY_RESPONSE | H: hallmark gene sets | 34/452 | 200/4383 | 0.01258 | 0.00927 |

was used to characterize transcriptome differences between tumor and normal samples. As shown in Fig 1B, we found 3,177 genes that were at least 1.5-fold up or down regulated (up = 1593, down = 1584) and had a p-value (Benjamini-Hochberg adjusted) less than or equal to 0.05. The heatmap in Fig 1C summarizes the dramatic differences in gene expression in normal and tumor samples.

*Malassezia* is proposed to promote tumor growth via the complement pathway, disruption of which activates inflammation [8, 15]. Gene set enrichment analysis (GSEA) can reveal pathway and disease phenotypes within sets of differentially expressed genes. GSEA of the set of genes differentially expressed between tumor and normal samples against the curated (C2), ontology (C5), and hallmark (H) gene sets at the molecular signatures database [16, 17], revealed enrichment for genes involved in the complement cascade, complement activation, and inflammatory response. Additionally, our differentially expressed genes are enriched for expected gene sets such as pancreatic cancer, epithelial mesenchymal transition, and KRAS signaling (Table 3).

There is an extensive literature of the linkage between the complement/inflammation genes and cancer, including PDAC, focusing primarily on the tumor microenvironment and how immune cells can affect tumor growth through the inflammatory process [18, 19]. Since each of these pathways were significantly enriched in our dataset (Table 3), we specifically examined which genes were differentially expressed, and their directionality. As expected, only a subset of the genes in each gene set were significantly differentially regulated: 64 of 200 for complement (Fig 2A), 52 of 200 for inflammatory (Fig 2B), and 104 of 200 for EMT (Fig 2C). Interestingly, overall expression of C3 was higher than C2 expression on both tumors and normals, but only C2 was significantly up-regulated in tumors (Fig 3).

We were particularly interested in the expression pattern of genes known to play a role in pancreatic cancer, complement, inflammation and EMT. Using the genesets described above, as well as a pancreatic cancer genesets (Gruetzmann pancreatic cancer up and Gruetzmann pancreatic cancer down [20], we looked for genes involve in multiple processes that indicate

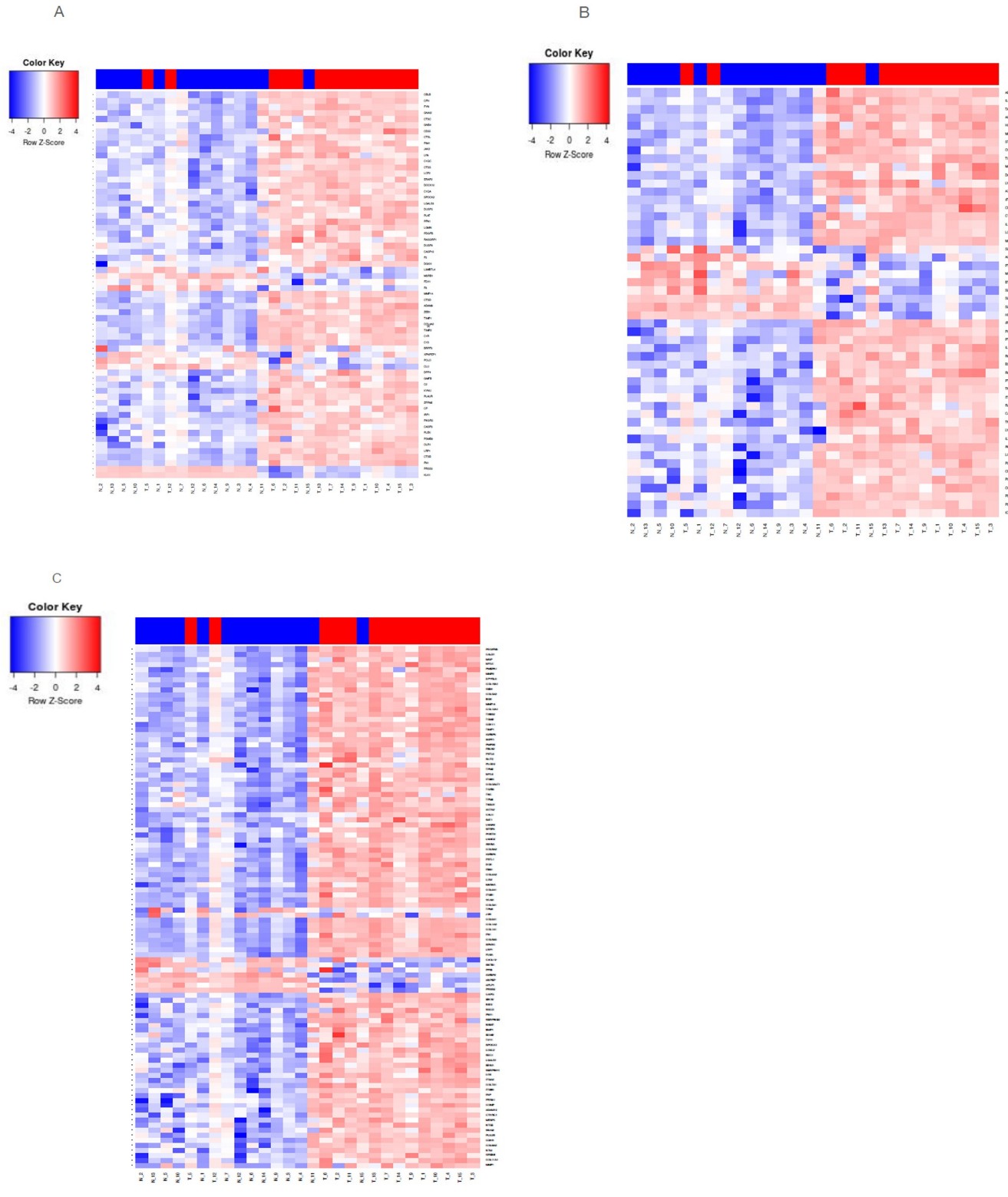

**Fig 2. Heatmaps of geneset genes that are differentially expressed between tumor (red bar) and normal (blue bar) PDAC samples.** (A) Hallmark Complement Geneset, the C2 gene is indicated by (*). (B) Hallmark Epithelial Mesenchymal Transition Geneset. (C) Hallmark Inflammation Response Geneset. (Ref = PMID 26771021).

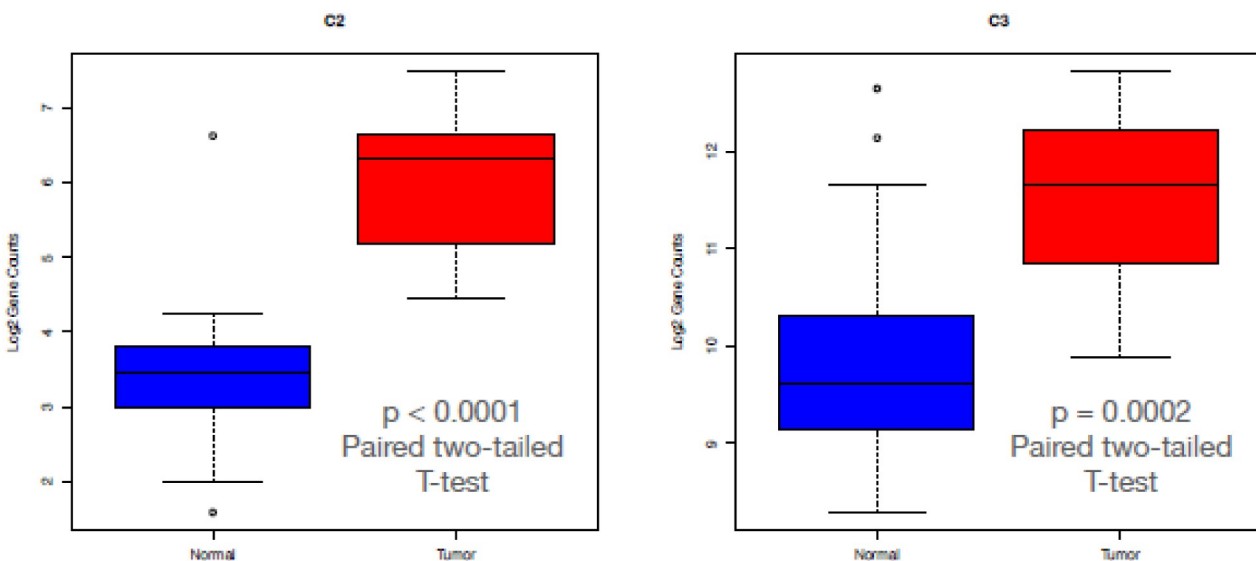

**Fig 3. Boxplots of log2 gene counts for C2 (left) and C3 (right) in normal samples (blue) and tumor samples (red).** The difference in C2 expression between normal (mean = 3.48; SD = 1.16) and tumor samples (mean = 16.01; SD 0.91) was significant (t (13) = -2.53 p < 0.0001). The difference in C3 expression between normal (mean = 9.92; SD = 1.31) and tumor samples (mean = 11.51; SD 0.85) was significant (t (13) = 5.14 p = 0.0002).

networks of influential genes using a CNET plot (Fig 4). This method showed that there were very few genes that overlapped the down regulated pancreatic cancer genes list and the complement pathway (1 gene), inflammatory response (1 gene), and EMT (3 genes) lists (Fig 4). However, we observed a number of genes that were up-regulated in pancreatic cancer and involved in both EMT (229 genes) and the complement pathways (14 genes), but a relative small number of inflammatory response genes (6 genes), all of which were up-regulated in our tumor samples. Interesting, PLAUR (urokinase plasminogen activator receptor) overlapped with all of these pathways, while FN1 (fibronectin 1) overlapped with both complement and EMT processes.

The upregulation and secretion of interleukin-33 (IL-33) in KRAS[G12D] PDAC subtype cells, and the interaction of these cells with intra-tumor fungi was reported. This resulted in the recruitment of $T_H2$ lymphoid cells and innate lymphoid cells (ILC2 cells) in the PDAC tumor micro-environment (TME). The intra tumor fungi identified were *Malassezia globoza* and *A Altermata*. Dectin-1 was activated by fungal components, which resulted in a dectin-1 mediated Src-Syk-CARD9 pathway activation. This in turn resulted in the secretion of IL-33, resulting in a pro-type 2 immune response and tumor progression in this PDAC subset. Reversal of IL-33 or anti-fungal treatment caused PDAC tumor regression [21]. The role of intra-tumor fungus is distinct from prior observation where a fungus was shown to activate the complement system [8].

Since our differentially expressed genes were also enriched for the Hallmark KRAS signaling pathway (Table 3), we also looked for overlaps between the KRAS signaling pathway geneset, and the other three sets used above. Fig 5 shows the CNET plot for the genesets alone, but the intersection of the geneset plus the PDAC genesets can be seen in S2 Fig in S1 Appendix. Our differentially expressed genes contain 5 genes which overlap both KRAS and complement pathways including PLAT, CFH, CTSS, DUS6, and PLAUR, (Fig 5), all of which are up-regulated in tumors. Interestingly, PLAUR and its ligand (urokinase plasminogen activator (uPA))

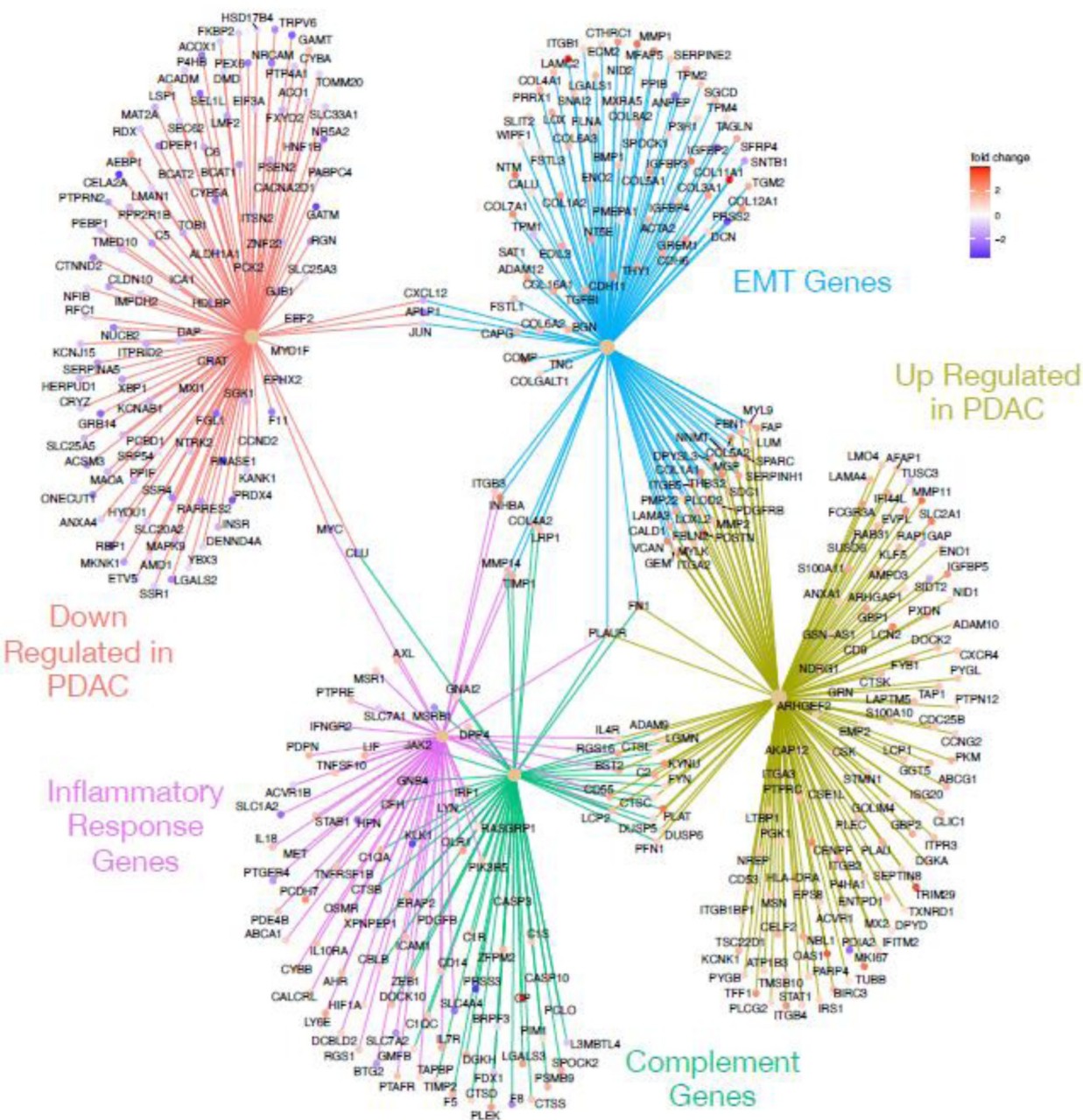

**Fig 4. CNET plot showing gene overlaps in enriched genesets.** Geneset enrichment analysis of genes differentially expressed between tumor and normal samples indicated enrichment for sets of genes up (olive) or down (orange) regulated in pancreatic cancer, the complement cascade (green), epithelial mesenchymal transition (blue), and inflammatory response (purple). Genes at the end spokes indicate differential expressed in our dataset that were enriched in each geneset, color of dots indicated whether the gene was up (red tones) or down (blue tones) regulated in tumors relative to normal samples [17, 21, 35].

overlaps with all of the tested geneset (PDAC genes, complement and KRAS pathways, inflammatory and EMT genes, Figs 4 and 5, S2 Fig in S1 Appendix), suggesting this may be a good drug target for the treatment of PDAC. On the surface of B cells, CR2 binds to CD19 and the tetraspanin molecule, and this complex is an important co-receptor in B cells. The

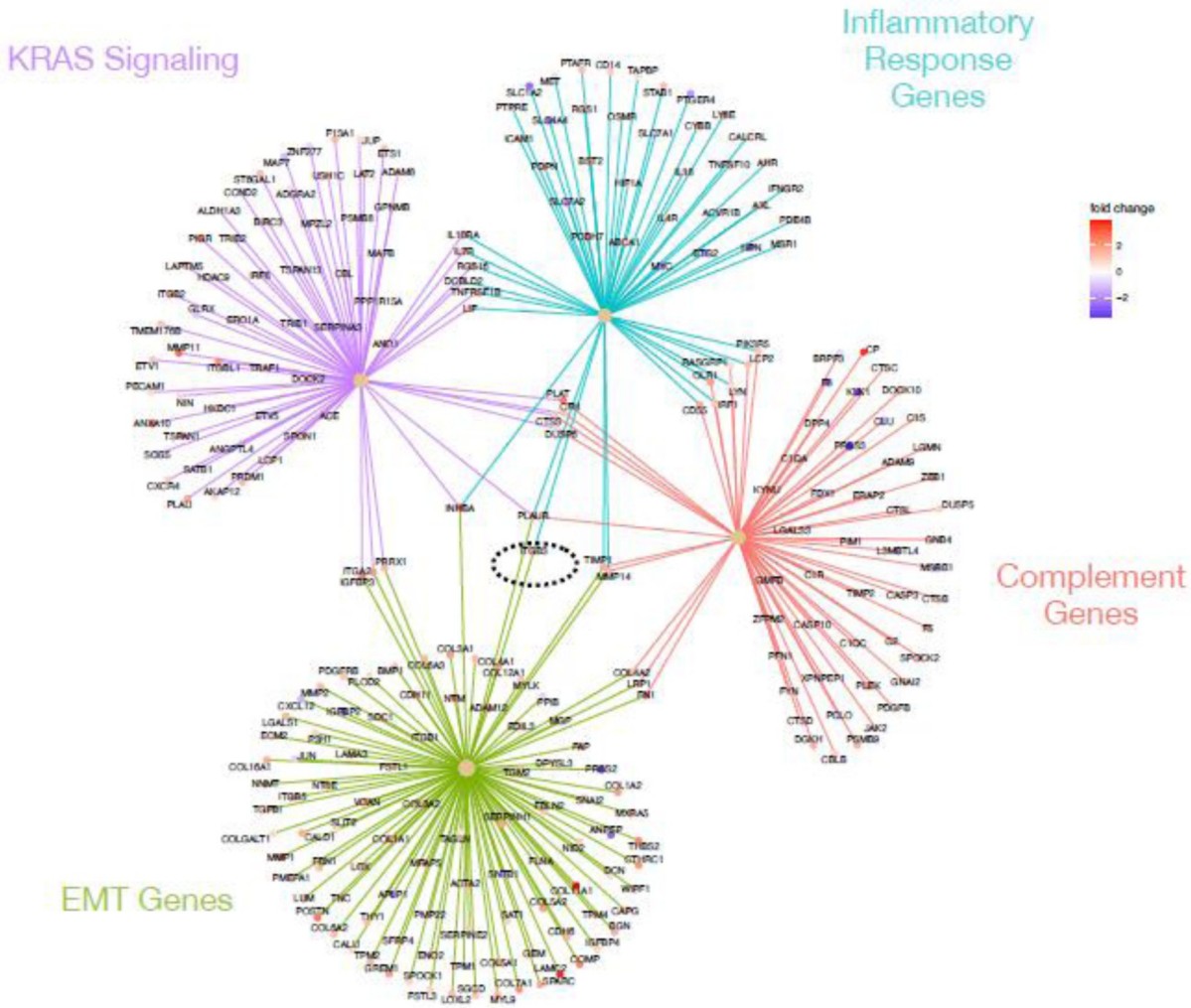

**Fig 5. CNET plot showing gene overlaps in enriched genesets.** Geneset enrichment analysis of genes differentially expressed between tumor and normal samples indicated enrichment for sets of genes involved in KRAS signaling (purple), inflammatory response (turquoise), complement cascade (red), and epithelial mesenchymal transition (green). Black dashed circle indicates PLAUR, which connects to all four genesets. Genes at the end spokes indicate differential expressed in our dataset that were enriched in each geneset, color of dots indicated whether the gene was up (red tones) or down (blue tones) regulated in tumors relative to normal samples [17, 21, 35].

complement receptor 2 (CR2/CD21) is a part of the complement pathway which activates C5 and the membrane attack complex (MAC). C3dg, a cleavage product of C3, can increase the complement receptor 2(CR2/CD21) binding to the B Cell Receptor (BCR) through co-ligation of C3dg/Ag complexes. CR2-mediated stimulation of peripheral B cell subpopulations demonstrated that CR2 can amplifying BCR signal transduction in the B-2 cells, but not B-1 cells. Furthermore, anti-IgM/C3dg conjugates did not stimulate B cells from CR2-deficient or CD19-deficient mice [22].

Human uPAR is a 55–60 kDa glycosylated protein that is bound to the outer lipid bilayer of the cell membrane by a glycosyl-phosphatidylinositol (GPI) anchor. Both uPA and/or uPAR allow malignant cells to dissolve the extracellular matrix (ECM) barriers and metastasize to distant sites due to the high proteolytic and migratory potential. UPAR is also believed to interact with cancer-associated intracellular signal-transduction pathways regulating other

tumor related pathways, including, proliferation, survival, migration, invasion, metastasis, angiogenesis, and epithelial–mesenchymal transition (EMT). In the paper they showed a graph of uPAR gene expression profile in human cancer. UPAR expression was highly overexpressed in PDAC [23].

We also found over-expression of Galectin-3 (Gal-3) and Dectin-1 in the tumor samples compared to the normal samples (Fig 6). Galectins are a group of conserved proteins with the ability to bind β-galactosides through certain carbohydrate-recognition domains (CRD). Gal-3 is structurally unique as it contains a C-terminal CRD linked to an N-terminal protein-binding domain, being the only chimeric galectin. Gal-3 also contributes in several ways to inflammation and to the innate immune response. Gal-3 is a multifunctional protein with roles in tumor cell adhesion, proliferation, differentiation, angiogenesis, metastasis, and apoptosis [24]. One group demonstrated that Gal-3 was highly expressed in human PDAC samples: 83 of 125 (66.4%), while only 32 of 108 (29%) of non PDAC pancreatic samples were positive for Gal-3 high expression. By using K-RAS mutated pancreatic cell lines genetically engineered to express high or low levels of Gal-3, they demonstrated that down regulation of Gal-3 decreased PDAC cell proliferation and invasion *in vitro* and reduced tumor volume in a mouse model. Conversely, transfection of Gal-3 cDNA into low-level Gal-3 PDAC cells, increased K-Ras activity. Gal-3 was found to bind to K-Ras and maintained K-Ras activity [25]. In another paper, translocation of Gal-3 from the nucleus to the cytoplasm or plasma membrane led to K-Ras stabilization, and a decrease in the K-Ras suppressor miRNA let-7 (let-7). Using a Gal-3-knockout mouse embryonic fibroblasts (MEFs) model, they demonstrated that Gal-3 down regulated let-7 which resulted in increased expression of K-Ras [26].

Dectin-1 recognizes β-glucans with its carbohydrate recognition domains (CRD) and transduces signals through its immune-receptor tyrosine-based activation motif (ITAM)-like motif in the cytoplasmic domain [27]. A recent study found that Dectin-1 is highly expressed on macrophages in pancreatic ductal adenocarcinoma (PDAC). Dectin-1 signaling resulted in

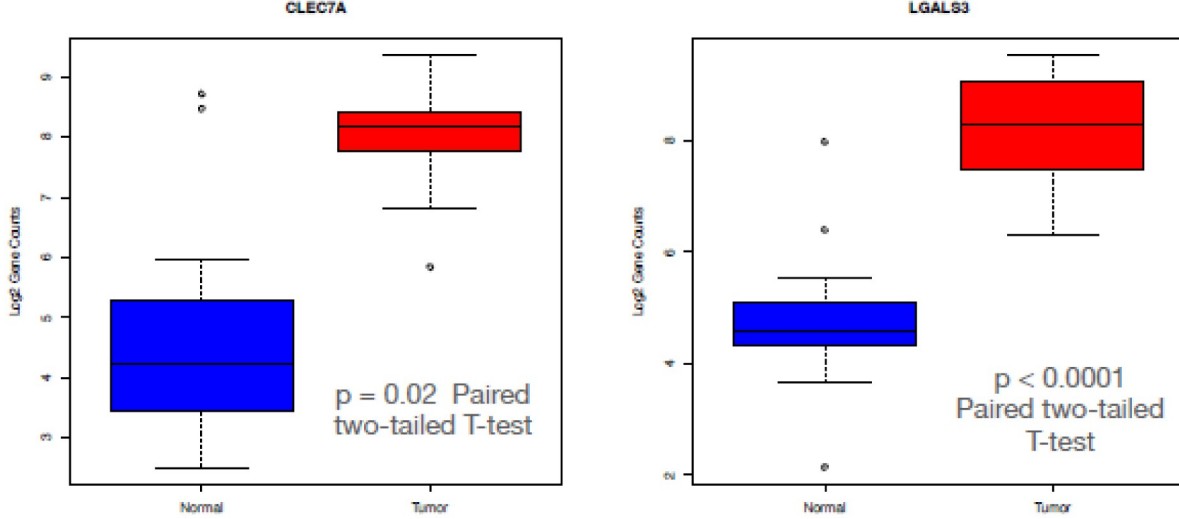

**Fig 6. Boxplots of log2 gene counts for CLEC7A (DECTIN-1 (left)) and LGALS3 (Galectin-3 (right)) in normal samples (blue) and tumor samples (red).** The difference in CLEC7A expression between normal (mean = 72.85; SD = 133.99) and tumor samples (mean = 288.16; SD = 142.84) was significant (t (13) = 3.86 p = 0.002). The difference in LGALS3 expression between normal (mean = 43.80; SD = 62.85) and tumor samples (mean = 358.13; SD 206.68) was significant (t (13) = 6.43 p < 0.0001).

PDAC progression, while Dectin-1 deletion or blockade of its downstream signaling decreased tumor growth. They showed that Syk phosphorylation was a downstream signal for Dectin-1 [28]. Furthermore the recent paper describing the effect of the fungus on PDAC subset (Kras G12D) via an alternative pathway (the IL-33-ILC2/$T_H$2 axis) is activated by a dectin-1 mediated Src-Syk-CARD9 pathway [21].

## Discussion

We explored the TME in the samples and found overexpression of C2, Dectin-1, and Galectin-3 represented in box plot analysis. There was no literature about the role of C2 in pancreatic cancer, but there was data showing C2 has a role in amplifying BCR signal transduction in each subpopulation of B-2 cells, but not B-1 cells, as noted above. Dectin-1 and Galectin-3 can augment PDAC progression as noted above. These two molecules can propagate the growth of the PDAC in vivo and mice models, but this must be looked at in the context of a multi molecular array, some inhibiting and some promoting PDAC growth.

We also compared the gene expression profiles of PDAC patient tumor samples to the normal sample and found several differential expressed genes. These genes were enriched for genes that are known to be involved in the complement cascade and inflammation, as well as genes involved in epithelial-mesenchyme transition, KRAS signaling, and known PDAC genes. From this we were able to identify one gene (PLAUR) that sits at the intersection of these pathways and may be a potential target for PDAC treatment. In addition, we found a handful of genes that represent additional attractive drug targets because they overlap subsets of complement, inflammation, KRAS and EMT pathways. Additionally, these results suggest that the TME contributes to PDAC progression by activation of these critical pathways. Examples of clinical correlation between PDCA prognosis and these genes include: 1) A correlation was noted between the overall survival of the pancreatic cancer patients with plasma Galectin-3 levels (n = 21). The high-galectin-3 group (n = 10) showed a poorer prognosis than the low-galectin-3 group (n = 11) in pancreatic cancer patients (p = 0.006) [29], and 2), PLAU upregulation was associated with the basal type of PDAC, and an increased PLAU protein expression was associated with poor prognosis (P = 0044), and within the basal subtype, the clinical outcome in the high PLAU expression group is significantly worse than the low PLAU expression group (P = 0.018) [30].

We did not have the data on the clinical outcome of the patients. Thus no biological correlative analysis was performed. Unlike the Aykut B et al [8] and Alam A et al [21] groups we noted the fungus *Malassezia* in the pancreatic cancer and normal control in relatively the same proportions. However, despite the fact that there was no difference in *Malassezia* in tumors versus normal tissue, there was a significant difference in expression of a number of genes related to inflammation, infection and EMT, suggesting other causes of the differences such as a the tumor itself as an example. This was a single institution study, with only 14 samples from the Hospital Tissue Bank. It was felt that tools like DeSeq2 and EdgeR are specifically designed to analyze RNA-seq counts, even with low sample sizes. They have consistently identified differential expression and pathways in previous cases with smaller sizes with unpaired samples. Our application of these tools successfully detected differentially expressed genes even after multiple hypothesis correction. However, we acknowledge that this may not have enough power to generalize the results. Based on our data we have no way of determining if this is a reaction to a fungus or the tumor itself. In summary, we have demonstrated the interaction between proteins involved in inflammation can also have a role in propagation of growth in pancreatic cancer growth.

## Methods

### RNA isolation and sequencing

Total RNA was extracted from FFPE slices using the RNeasy FFPE kit (Qiagen) and the manu-facture's protocol. Synthesis of cDNA and library preparation were performed using the SMARTer Universal Low Input RNA Kit for Sequencing (Clontech) and the Ion Plus Frag-ment Library Kit (ThermoFisher) as previously described [10, 31, 32]. Sequencing was per-formed using the Ion Proton S5/XL systems (ThermoFisher) in the Analytical and Translational Genomics Shared Resource at the UNM Comprehensive Cancer Center. RNA sequencing data is available for download from the NCBI BioProject database using study accession number PRJNA940178.

### Data analysis

Prior to alignment, non-human RNA-seq reads were identified and removed from analysis using the kraken2 taxonomic sequence classification system against a library containing human, fungal, bacterial and viral genomes [11–13] and final genera level abundances were calculated using Bracken (v2.5.0, [14]. High-quality, trimmed reads classified as human, were aligned to GRCh38 (hg38) using tmap (v5.10.11). Exon counts were calculated using HTseq (v0.11.1, [33] against a BED file containing non-overlapping exons from UCSC genome hg38 and gene counts were generated by summing counts across exons. Samples were normalized for library size using DEseq2 (v1.34.0, [34]) and low expressing genes were excluded from the final analysis using a filtering threshold of 20 reads in 9 samples. Multi-dimensional scaling (MSC) was performed using plotMDS from the limma package (v3.50.0) and the base stats package was used for unsupervised heirachrical clustering [35]. EdgeR (v 3.36.0 was used for the differential expression (crosswise comparison of the groups) using the glm method with an adjusted p-value (method = BH) of cutoff of 0.05 and requiring a minimum fold change of 1.5. ClusterProfiler (v 4.2.1), was used to analyze the differentially expressed genes for gene set enrichment (GSEA) against the Molecular Signatures Database and visualize differentially expressed genes in gene sets of interest [16, 17, 36]. Default values were used at all points [37]. For additional details see [10, 33]. All analysis was done using R version 4.1.2.

### Ethics statement

We performed studies on paraffin embedded pancreatic cancer biopsies using pancreatic can-cer and their normal tissue counterpart samples from the UNM Tissue Repository. These sam-ples were anonymized and thus we obtained no clinical data from these samples. The UNM Human Research Protection Program (HRCC) reviewed our submission on 12/6/2020. They determined that informed consent and HIPAA authorization addendums were not applicable to this study. We thus obtained no verbal or written consent.

## Supporting information

**S1 Appendix. S1 Fig**. The heatmap summarizes the supervised clustering and differential gene expression analysis comparing the normal samples to the tumor samples. Dendrograms show the unsupervised hierarchical clustering of samples(top) and genes (left side). **S2 Fig**. CNET plot showing gene overlaps in enriched genesets. Geneset enrichment analysis of genes differentially expressed between tumor andnormal samples indicated enrichment for sets of genes involved in KRAS signaling (purple), inflammatory response (blue), complement cas-cade (green),and epithelial mesenchymal transition (turquoise). Figure also includes genes known to up (orange) or down (olive) regulated in PDAC. Black dashed circle indicates

PLAUR, which connects to all 5 of the 6 genesets (not connected to down in PDAC). Genes at the end spokes indicate differential expressed inour dataset that were enriched in each geneset, color of dots indicated whether the gene was up (red tones) or down (blue tones) regulated in tumors relative to normal samples [17, 21, 36].
(PDF)

## Author Contributions

**Conceptualization:** Shashank Cingam, Ian Rabinowitz.

**Data curation:** Kathryn J. Brayer, Joshua A. Hanson, Cathleen Martinez.

**Formal analysis:** Kathryn J. Brayer, Scott A. Ness.

**Funding acquisition:** Scott A. Ness, Ian Rabinowitz.

**Investigation:** Kathryn J. Brayer, Cathleen Martinez, Scott A. Ness, Ian Rabinowitz.

**Methodology:** Kathryn J. Brayer, Scott A. Ness.

**Resources:** Scott A. Ness.

**Supervision:** Joshua A. Hanson, Scott A. Ness.

**Writing – original draft:** Kathryn J. Brayer, Shashank Cingam, Ian Rabinowitz.

**Writing – review & editing:** Kathryn J. Brayer, Scott A. Ness, Ian Rabinowitz.

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
