## [Decision Letter · Decision Letter 0]

2 May 2023

PONE-D-23-09005The immune response to a fungus in pancreatic cancer samples

PLOS ONE

Dear Dr. Rabinowitz,

Thank you for submitting your manuscript to PLOS ONE. After careful consideration, we feel that it has merit but does not fully meet PLOS ONE’s publication criteria as it currently stands. Therefore, we invite you to submit a revised version of the manuscript that addresses the points raised during the review process.

Following are some of my own suggestions for making changes to the paper:

14 samples, even though they are paired might be too low of a number, are you sure you have enough power to do the study? You could also study genes from the TCGA database besides your own data set and the ones from Gruetzmann dataset.Line 53: how much longer specifically did those patients live who had a richer gut microbial community.Lines 103-106: 3,177 genes is quite a large number to find. Did you use Benjamini-Hochberg correction? The number of false positive genes might be too high.Line 250: Why did you not use DESeq2?Line 254; in what way did you use clusterProfiler? Since you used these tools to generate results, it is essential to document their use.Line 176-177: the term “evolutionarily conserved” is a very poor choice of terms. I suggest not using it, since it is an oxymoron. In a broad sense, evolution means change, but conservation denotes stasis. The term evolutionary conservation implies something changing while standing still.Line 195: “β-glucans is a major cell wall component of fungi”, this is an orphan sentence, why is this statement relevant to your findings?Lines 205-206: please add a reference supporting the claim that C2 has a role in BCR signal transduction.Line 219: do you have clinical results from similar studies?Lines 222-227: you state that there was no difference in the *Malassezia* between tumor and non-tumor samples. Why then do you think that altering the microbial flora around and in the tumor will have a therapeutic effect?

We look forward to receiving your revised manuscript.

Kind regards,

Matthew Cserhati, Ph.D

Academic Editor

PLOS ONE

Journal Requirements:

“This research was partially supported by UNM Comprehensive Cancer Center Support Grant NCI P30CA118100 (SN, KB,& IR) and the Analytical and Translational Genomics Shared Resource, which receives additional support for the State of New Mexico R01DE023222 (SN, KB), and by Department of Defense grant: Defense Threat Reduction Agency https://www.dtra.mil/HDTRA12010015 (SN,KB).”

“This research was partially supported by the University of New Mexico Comprehensive Cancer Center Support Grant NCI University of New Mexico Cancer Research & Treatment Center P30CA118100(IR) https://unmhealth.org/cancer/research/sharedresources/

and the Analytical and Translational Genomics Shared Resource, which

receives additional support for the State of New Mexico.(SN,KB) R01DE023222

https://app.dimensions.ai/details/grant/grant.2491524, U2CCA252973

https://www.ncbi.nlm.nih.gov/pmc/articles/PMC9223088/(SN,KB) and University of

New Mexico Comprehensive Cancer Center Support Grant NCI University of New Mexico Cancer Research & Treatment Center P30CA118100(SN,KB)

https://unmhealth.org/cancer/research/shared-resources/and by Department of Defence grant: Defence Threat Reduction Agency https://www.dtra.mil/HDTRA12010015.(SN,KB).

The funders had no role in study design, data collection and analysis, decision to

publish, or preparation of the manuscript.”

“I have read the journal's policy and the authors of this manuscript have the following competing interests: NONE”

6. Please upload a new copy of Figures 1, 2, 3, 4, 5 and 6 as the detail is not clear. Please follow the link for more information: https://blogs.plos.org/plos/2019/06/looking-good-tips-for-creating-your-plos-figures-graphics/

7. Please include a new copy of Table 2 in your manuscript; the current table is difficult to read. Please follow the link for more information: https://blogs.plos.org/plos/2019/06/looking-good-tips-for-creating-your-plos-figures-graphics/

8. Please include your tables as part of your main manuscript and remove the individual files. Please note that supplementary tables should be uploaded as separate "supporting information" files.

Reviewers' comments:

Reviewer's Responses to Questions

**Comments to the Author**

1. Is the manuscript technically sound, and do the data support the conclusions?

Reviewer #1: Partly

Reviewer #2: Partly

2. Has the statistical analysis been performed appropriately and rigorously? 

Reviewer #1: Yes

Reviewer #2: I Don't Know

3. Have the authors made all data underlying the findings in their manuscript fully available?

Reviewer #1: Yes

Reviewer #2: Yes

4. Is the manuscript presented in an intelligible fashion and written in standard English?

Reviewer #1: Yes

Reviewer #2: Yes

5. Review Comments to the Author

Reviewer #1: Although the points made by the authors contribute to our understanding of pancreatic cancer. But I think this article needs to be overhauled before it might be suitable for publication on PLOS ONE.This study used RNA sequencing to compare 14 paired tumor and normal (tumor adjacent) pancreatic cancer samples and found Malassezia RNA in both the PDAC and normal tissues. The authors claimed to find that the presence of Malassezia was not correlated with tumor growth, but involved in tumor progression.They believe that the evidence that fungi promote tumor progression is that genes related to inflammation and immune response are upregulated in tumor tissue. This doesn't seem very convincing. I recommend authors to conduct In vitro and animal experiments to verify the results obtained by RNA sequencing.In addition, the clarity of the figure cannot be seen by the reader (e.g. Figures 1, 2, ).

Reviewer #2: Title: The immune response to a fungus in pancreatic cancer samples

This paper presents a study to include pancreatic ductal adenocarcinoma RNA-seq expression data across 14 paired samples (Tumor vs Normal) to assess the performance of C3 and Malassezia and associated genes using differential expression analysis and gene set enrichment analysis. Authors showed the activation of the complement cascade pathway and inflammation could be involved in pro PDAC growth. However, there are some questions as outlined below.

1- The PDAC RNA-seq summary (Table1) is based on normalized count data and after filtration? Please add more details at the caption along with total number of genes remain after filtration.

2- I assume authors considered the same format of data for all the analysis including filtered normalized data.

a. As an example, for Figure 3 & 6, it was mentioned log2 count expression. So, do authors keep working on log2 normalized RNA-seq (after filtration)? Please clarify this part.

3- Figure 1A shows the MDS analysis. The heterogeneity across tumor samples can be related to the tumor sub-types. Is there any data information about the sub-type of PDAC data? If yes, that’s great to include sub-type info in MDS analyses.

4- Table 2 represents the pathway enrichment analysis while the column between pvalue and qvalue is not clear. Please keep consistency about the multiple test correction approach and its notation, qvalue, adjusted p-value, or BH?

5- Method section needs to be improved

a. edgeR package includes different normalization method. Which one was applied for data?

b. MDS or PCA was applied?

c. I couldn’t follow why and where DESeq2 was applied?

d. How authors include the paired sample differentially expressed analysis using edgeR package?

e. Please add other analysis were used in the manuscript including GSEA

f. I assume heatmaps show unsupervised clustering (e.g., SFig1)?

g. Boxplots (Fig 3 & 6) show considerable differences between tumor and normal. It is great to choose an appropriate statistical method and add the p-value on each figure.

h. Add the type of multiple test correction approach that was applied across manuscript and potential cut-off (for all analysis).

6- Authors did findings based on only one PDAC cohort. Did authors try any other cohorts using the same analysis to see if it is possible to generalize their findings?

a. What are the main limitations of this research study? This cohort has few samples. Is there any treatment response data for the samples or any other molecular data (DNA)?

b. Also is it possible to share how can get access to this cohort?

7- Figures have low resolutions and need to be improved.

a. As an example, the heatmaps are not clear.

b. Please add on the caption, the number of genes included in heatmaps or volcano, and MDS plot

c. Y-axis at the volcano plot shows the -log10 p-value (where the p-value shows adjusted p-value or BH?). The figure needs to be improved.

i. Y-axis can be -log10 p-value

ii. Colored up- and down-regulated genes using qvalue (or BH) and fold change

iii. Vertical lines for fold change cut-off and vertical line for p-value

6. PLOS authors have the option to publish the peer review history of their article (what does this mean?). If published, this will include your full peer review and any attached files.

Reviewer #1: No

Reviewer #2: No

---

## [Author Response · Author response to Decision Letter 0]

1 Aug 2023

Response to Reviewers

1. 14 samples, even though they are paired might be too low of a number, are you sure you have enough power to do the study? You could also study genes from the TCGA database besides your own data set and the ones from Gruetzmann dataset.

Answer: 

1. A UNM statistician felt that tools like DeSeq2 and EdgeR are specifically designed to analyze RNA-seq counts, even with low sample sizes. They have consistently identified differential expression and pathways in previous cases with smaller sizes with unpaired samples. Our application of these tools successfully detected differentially expressed genes even after multiple hypothesis correction, which is statistically sound.

a. We used the Grützmann data set as referenced in ref (20) 

 Manuscript' line 140

 'Revised Manuscript with Track Changes Line 150

b We used EdgR ref (34, 35)

 Manuscript line 268 

 ‘Revised Manuscript with Track Changes line 280

c We used DeSeq2 ref (34)

 Manuscript line 265 

 ‘Revised Manuscript with Track Changes line 282

2. Line 53: how much longer specifically did those patients live who had a richer gut microbial community.

Answer: “. Patients with high diversity and low diversity had overall survival of 9.66 vs. 1.66 years respectively using univariate Cox proportional hazard models”. This has been added to the manuscript. 

Manuscript' Line 56

'Revised Manuscript with Track Changes Line 56

3. Lines 103-106: 3,177 genes is quite a large number to find. Did you use Benjamini-Hochberg correction? The number of false positive genes might be too high.

Answer: Yes. We did use the Benjamini-Hochberg (BH) correction. 

“As shown in Figure 1B, we found 3,177 genes that were at least 1.5-fold up or down regulated (up = 1593, down = 1584) and had a p-value (Benjamini-Hochberg adjusted) less than or equal to 0.05.”

Manuscript' Line 111: 

'Revised Manuscript with Track Changes: Line 112: 

4. Line 250: Why did you not use DESeq2?

Answer: We used EdgeR which in our opinion is a perfectly acceptable alternative tool. However we also used DESeq2. We realized that the methods section says DESeq, and not DESeq2. We corrected the typo to accurately reflect our use of DESeq2 and not DESeq. 

Manuscript: Line 265

'Revised Manuscript with Track Changes Line 281

5. Line 254; in what way did you use clusterProfiler? Since you used these tools to generate results, it is essential to document their use.

Answer: ClusterProfiler, an R package, was used to analyze the differentially expressed genes for gene set enrichment (GSEA) against the Molecular Signatures Database and visualize differentially expressed genes in gene sets of interest (16, 17, 36). Default values were used at all points.

Manuscript Line 270: 

'Revised Manuscript with Track Changes: Line 290 

6. Line 176-177: the term “evolutionarily conserved” is a very poor choice of terms. I suggest not using it, since it is an oxymoron. In a broad sense, evolution means change, but conservation denotes stasis. The term evolutionary conservation implies something changing while standing still. 'Revised Manuscript with Track Changes

Answer: We removed the word “evolutionary” from the manuscript

Manuscript Line 187

'Revised Manuscript with Track Changes Line 198

7. Line 195: “β-glucans is a major cell wall component of fungi”, this is an orphan sentence, why is this statement relevant to your findings?

Answer: We removed the sentence from the manuscript. 

Manuscript Line 203

'Revised Manuscript with Track Changes Line 217

8. Lines 205-206: please add a reference supporting the claim that C2 has a role in BCR signal transduction.

Answer: We placed a reference (22) with some discussion supporting the claim that C2 has a role in BCR signal transduction. 

Manuscript Line 177 ref (22), line 170 (discussion)

'Revised Manuscript with Track Changes Line 147 reference (22), 140 (discussion)

9. Line 219: do you have clinical results from similar studies?

Answer: We added data of 2 clinic papers supporting the effect of galectin-3 and PLAU on PDCA prognosis. 

Examples of clinical correlation between PDCA prognosis and these genes include: 1) A correlation was noted between the overall survival of the pancreatic cancer patients with plasma Galectin-3 levels (n=21). The high-galectin-3 group (n=10) showed a poorer prognosis than the low-galectin-3 group (n=11) in pancreatic cancer patients (p=0.006). (ref 29), and 2), PLAU upregulation was associated with the basal type of PDAC, and an increased PLAU protein expression is associated with poor prognosis (P=0044), and within the basal subtype, the clinical outcome in the high PLAU expression group is significantly worse than the low PLAU expression group (P=0.018). (ref 30).

Manuscript Line 229

'Revised Manuscript with Track Changes Line 240

10. Lines 222-227: you state that there was no difference in the Malassezia between tumor and non-tumor samples. Why then do you think that altering the microbial flora around and in the tumor will have a therapeutic effect?

Answer: We removed the sentence “The role of the microbiome, including fungi is being recognized as having a significant role in the pathophysiology and progression of PDAC, and we believe solving how to alter the flora around and in the tumor can add another dimension to treating this disease” We are not going to comment about the microbiome with respect to outcome of PDCA, as our data does not address this issue. 

Manuscript Line 240

'Revised Manuscript with Track Changes Line 254

Journal Requirements

1. We have adapted the PLOS ONE’s style requirements

2. We have removed funding acknowledgments from the manuscript, and have added: “The funders had no role in study design, data collection and analysis, decision to publish, or preparation of the manuscript” to the cover letter

3. We have added: “I have read the journal's policy and the authors of this manuscript have the following competing interests: NONE” in the cover letter.

4. In your Data Availability statement, you have not specified where the minimal data set underlying the results described in your manuscript can be found. PLOS defines a study's minimal data set as the underlying data used to reach the conclusions drawn in the manuscript and any additional data required to replicate the reported study findings in their entirety. All PLOS journals require that the minimal data set be made fully available. For more information about our data policy, 

Genomics Shared Resource at the UNM Comprehensive Cancer Center. RNA sequencing data is available for download from the NCBI BioProject database using study accession number PRJNA940178. However, this will be made available to review once the paper has been accepted.

We have added the full ethics statement in the Methods section

6. Please upload a new copy of Figures 1, 2, 3, 4, 5 and 6 as the detail is not clear. Please follow the link for more information

We improved the figures 1,2,3,4,5, and 6 firstly by taking them out their ”boxes”, and displayed them one by one, in the manuscript. We will also send a copy of the figures independently, which have better visualization characteristics. Figure 2 C is still not perfect but is fine in the attached copies of the figures.

7. Please include a new copy of Table 2 in your manuscript; the current table is difficult to read.

We have corrected table 2 (now table 3)

8. Please include your tables as part of your main manuscript and remove the individual files. Please note that supplementary tables should be uploaded as separate "supporting information" files.

We put the three tables in the main manuscript and the Heat map and CNET plots supplementary figures in the supporting information.

Additional Reviewers Questions

1- The PDAC RNA-seq summary (Table1) is based on normalized count data and after filtration? Please add more details at the caption along with total number of genes remain after filtration. This table shows raw, unfiltered and un-normalized sequencing data statistics. The caption was edited to include additional information to clarify this point as well as how the information was determined. 

2- I assume authors considered the same format of data for all the analysis including filtered normalized data. 

a. As an example, for Figure 3 & 6, it was mentioned log2 count expression. So, do authors keep working on log2 normalized RNA-seq (after filtration)? Please clarify this part. Updated

3- Figure 1A shows the MDS analysis. The heterogeneity across tumor samples can be related to the tumor sub-types. Is there any data information about the sub-type of PDAC data? If yes, that’s great to include sub-type info in MDS analyses. No

4- Table 2 represents the pathway enrichment analysis while the column between p value and q value is not clear. Please keep consistency about the multiple test correction approach and its notation, qvalue, adjusted p-value, or BH? Fixed. Also fixed to indicate definition of GeneRatio and BgRatio

5- Method section needs to be improved Updated

a. edgeR package includes different normalization method. Which one was applied for data? Clarified 

b. MDS or PCA was applied? Clarified

c. I couldn’t follow why and where DESeq2 was applied? Clarified

d. How authors include the paired sample differentially expressed analysis using edgeR package? Corrected misleading terminology. 'Paired' was replaced with 'matched' to indicate that both the tumor and normal samples came from the same patient, however to analysis was not a paired analysis. 

e. Please add other analysis were used in the manuscript including GSEA

f. I assume heatmaps show unsupervised clustering (e.g., SFig1)? No, as stated in the figure legend this heatmap is a supervised heatmap. 

g. Boxplots (Fig 3 & 6) show considerable differences between tumor and normal. It is great to choose an appropriate statistical method and add the p-value on each figure. Added

h. Add the type of multiple test correction approach that was applied across manuscript and potential cut-off (for all analysis). Updated

6- Authors did findings based on only one PDAC cohort. Did authors try any other cohorts using the same analysis to see if it is possible to generalize their findings? No 

a. What are the main limitations of this research study? This cohort has few samples. Is there any treatment response data for the samples or any other molecular data (DNA)? This a small sample. The samples were anonymized, and thus there was no clinical data to review. There is PDCA prognosis data in humans for gal-3 and UPAR that is noted in the manuscript.

b. Also is it possible to share how can get access to this cohort? Unfortunately no

7- Figures have low resolutions and need to be improved.

a. As an example, the heatmaps are not clear. done

b. Please add on the caption, the number of genes included in heatmaps or volcano, and MDS plot done

c. Y-axis at the volcano plot shows the -log10 p-value (where the p-value shows adjusted p-value or BH?). The figure needs to be improved. done

i. Y-axis can be -log10 p-value

---

## [Decision Letter · Decision Letter 1]

16 Aug 2023

PONE-D-23-09005R1The immune response to a fungus in pancreatic cancer samplesPLOS ONE

Dear Dr. Rabinowitz,

Thank you for submitting your manuscript to PLOS ONE. After careful consideration, we feel that it has merit but does not fully meet PLOS ONE’s publication criteria as it currently stands. Therefore, we invite you to submit a revised version of the manuscript that addresses the points raised during the review process.

We look forward to receiving your revised manuscript.

Kind regards,

Hilary A. Coller

Academic Editor

PLOS ONE

Additional Editor Comments (if provided):

Thank you for your submission. The reviewers have major concerns about this manuscript that require your attention. In particular, the reviewers are concerned that changes in inflammation may not reflect the presence or absence of the fungus, but could be a response to the presence of the tumor itself. Please be sure to address this concern in your response.

Reviewers' comments:

Reviewer's Responses to Questions

**Comments to the Author**

1. If the authors have adequately addressed your comments raised in a previous round of review and you feel that this manuscript is now acceptable for publication, you may indicate that here to bypass the “Comments to the Author” section, enter your conflict of interest statement in the “Confidential to Editor” section, and submit your "Accept" recommendation.

Reviewer #1: All comments have been addressed

Reviewer #2: All comments have been addressed

2. Is the manuscript technically sound, and do the data support the conclusions?

Reviewer #1: Partly

Reviewer #2: Yes

3. Has the statistical analysis been performed appropriately and rigorously? 

Reviewer #1: I Don't Know

Reviewer #2: Yes

4. Have the authors made all data underlying the findings in their manuscript fully available?

Reviewer #1: Yes

Reviewer #2: Yes

5. Is the manuscript presented in an intelligible fashion and written in standard English?

Reviewer #1: Yes

Reviewer #2: Yes

6. Review Comments to the Author

Reviewer #1: (No Response)

Reviewer #2: Authors already addressed all the comments. However, this study does not enough power to generalize the results due to the limited number of samples.

7. PLOS authors have the option to publish the peer review history of their article (what does this mean?). If published, this will include your full peer review and any attached files.

Reviewer #1: No

Reviewer #2: No

---

## [Author Response · Author response to Decision Letter 1]

31 Aug 2023

Letter of Response to the academic editor and reviewers. 8/24/2023

PONE-D-23-09005R1

Case 08128318

PL#0N3_AR

ref:_00DU0Ifis._500PMn9N8:ref

Major Concern: The changes of inflammation may not reflect the presence or absence of the fungus but could represent the tumor itself.

Answer

Line 1 new line 1 tracked

We agree with this sentiment and have made changes to the manuscript.

a. We have changed the title of manuscript from:

“The immune response to a fungus in pancreatic cancer samples”

To:

“The inflammatory response of human pancreatic cancer samples compared to normal controls”

b. Line 250 new tracked version 250

We added this sentence in the discussion.

“Based on our data we have no way of determining if this is a reaction to a fungus or the tumor itself.”

Comments to Author

Answer

1. Both Reviewers said “all comments have been addressed”

2. Is the manuscript technically sound and do the data support the conclusion

a. Reviewer #1 said partly, and Reviewer #2 said yes

Answer

It is unclear what the “partly” refers to, but we have addressed a number of issues in this response letter. 

3. Has the Statistical analysis been performed appropriately and rigorously

Answer

Reviewer #1 said “I don’t know”, reviewer #2 said yes.

To answer Reviewer # 1: There is an overlap of question 3 and 6. 

For #3: As we have mentioned before: We have added a similar passage into the manuscript (in lines 244-250 new and 244-250 tracked versions) for question #6

This was a single institution study, and we got the samples from the Hospital Tissue Bank. A UNM statistician felt that tools like DeSeq2 and EdgeR are specifically designed to analyze RNA-seq counts, even with low sample sizes. They have consistently identified differential expression and pathways in previous cases with smaller sizes with unpaired samples. Our application of these tools successfully detected differentially expressed genes even after multiple hypothesis correction, which he believed is statistically sound. 

4. Data of findings of the manuscript fully available

Answer

Both the reviewers said yes

5. The manuscript was presented in an intelligible fashion and written in standard English

Answer

a. Both reviewers said yes

6. Review comments to the Author

a. Reviewer # 1. (No Response)

Reviewer # 2 said the study does not have enough power to generalize the results due to a limited number of samples

Answer

This issue is addressed on pages. 

Line 244-50 new and tracked lines 244-250 versions

This was a single institution study, with only 14 samples from the Hospital Tissue Bank. It was felt that tools like DeSeq2 and EdgeR are specifically designed to analyze RNA-seq counts, even with low sample sizes. They have consistently identified differential expression and pathways in previous cases with smaller sizes with unpaired samples. Our application of these tools successfully detected differentially expressed genes even after multiple hypothesis correction. However, we acknowledge that this may not have enough power to generalize the results.

Sincerely 

Ian Rabinowitz

---

## [Editor Report · Decision Letter 2]

4 Sep 2023

The inflammatory response of human pancreatic cancer samples compared to normal controls

PONE-D-23-09005R2

Dear Dr. Rabinowitz,

We’re pleased to inform you that your manuscript has been judged scientifically suitable for publication and will be formally accepted for publication once it meets all outstanding technical requirements.

Kind regards,

Hilary A. Coller

Academic Editor

PLOS ONE
---

## [Editor Report · Acceptance letter]

3 Oct 2023

PONE-D-23-09005R2 

The inflammatory response of human pancreatic cancer samples compared to normal controls 

Dear Dr. Rabinowitz:

I'm pleased to inform you that your manuscript has been deemed suitable for publication in PLOS ONE. Congratulations! Your manuscript is now with our production department. 

Kind regards, 

on behalf of

Dr. Hilary A. Coller 

Academic Editor

PLOS ONE